# Entanglement and teleportation between polarization and wave-like encodings of an optical qubit

Demid V. Sychev [1,2], Alexander E. Ulanov [1,3], Egor S. Tiunov[1,3], Anastasia A. Pushkina[1,4], A. Kuzhamuratov[1,3], Valery Novikov[1,3] & A.I. Lvovsky[1,4,5,6]

Light is an irreplaceable means of communication among various quantum information processing and storage devices. Due to their different physical nature, some of these devices couple more strongly to discrete, and some to continuous degrees of freedom of a quantum optical wave. It is therefore desirable to develop a technological capability to interconvert quantum information encoded in these degrees of freedom. Here we generate and characterize an entangled state between a dual-rail (polarization-encoded) single-photon qubit and a qubit encoded as a superposition of opposite-amplitude coherent states. We furthermore demonstrate the application of this state as a resource for the interfacing of quantum information between these encodings. In particular, we show teleportation of a polarization qubit onto a freely propagating continuous-variable qubit.

[1] Russian Quantum Center, 100 Novaya St., Skolkovo, Moscow, Russia 143025. [2] Department of Theoretical Physics, Moscow State Pedagogical University, M. Pirogovskaya Street 29, Moscow, Russia 119991. [3] Moscow Institute of Physics and Technology, Dolgoprudny, Russia 141700. [4] Institute for Quantum Science and Technology, University of Calgary, Calgary, AB T2N 1N4, Canada. [5] P. N. Lebedev Physics Institute, Leninskiy prospect 53, Moscow, Russia 119991. [6] Clarendon Laboratory, University of Oxford, Parks Road, Oxford OX1 3PU, UK. These authors contributed equally: Demid V. Sychev, Alexander E. Ulanov, Egor S. Tiunov. Correspondence and requests for materials should be addressed to A.I.L. (email: lvov@ucalgary.ca)

Different physical systems with the potential for quantum processing and storage can be roughly classified into two categories. Some systems, such as single atoms, quantum dots, superconducting circuits, or color centers, have non-equidistant energy level structures, from which one can select a pair of levels that can serve as a qubit. For other systems, e.g., atomic ensembles, optical or microwave cavities, and opto-mechanical membranes, the energy level structure is inherently equidistant, and therefore analogous to that of the harmonic oscillator. In these systems, it may be more beneficial to encode quantum information in continuous degrees of freedom, such as the position and momentum.

Because different quantum systems are more suitable for performing different tasks, a technology for coherent and loss-free exchange of quantum information among them is essential for efficient integrated quantum information processing[1]. A natural mediator for such exchange is the electromagnetic field, which is the only quantum system capable of carrying quantum information over significant distances. Fortunately, this field is capable of coupling efficiently to both qubit-like and harmonic-oscillator-like systems through its own discrete-[2] and continuous-variable (CV)[3] degrees of freedom.

The most common discrete-variable (DV) approach to encoding quantum information in an optical wave is the dual-rail qubit: a single photon occupying one of two orthogonal modes corresponds to logical 0 or 1. These two modes can correspond, for example, to the horizontal $|H\rangle$ and vertical $|V\rangle$ polarizations. In the CV domain, a qubit can be encoded as a superposition of coherent fields of opposite phases, $|\gamma\rangle$ and $|-\gamma\rangle$, with the amplitude $\gamma$ being high enough to ensure sufficient orthogonality of these states[4,5]. An alternative encoding basis in CV consists of "Schrödinger cat" states[6–9] $|\Theta_\pm\rangle = \mathcal{N}_\pm(|\gamma\rangle \pm |-\gamma\rangle)$, where $\mathcal{N}_\pm = 1/\sqrt{2 \pm 2e^{-2\gamma^2}}$ is the normalization factor.

A missing central piece in the technology of electromagnetic coupling of different physical systems is a method for inter-converting between DV and CV encodings of the electromagnetic qubit. Important achievements towards this challenge have been reported in 2014 by two groups[10,11]. They constructed an entangled state between the CV qubit and a "single-rail" DV qubit in which the logical value is encoded in a single photon being present or absent in a certain mode. Subsequently, this state has been employed as a resource for rudimentary quantum tele-portation between these qubits[12].

However, the single-rail encoding of the qubit is much less common in practical quantum optical information processing than its dual-rail counterpart. This is because single-rail encoding complicates single-qubit operations[13] and also enhances the qubit measurement errors associated with optical losses and inefficient detection. In this case, it may be more beneficial to create a two-mode state of the form

$$|R\rangle = \alpha|H\rangle|\Theta_+\rangle + \beta|V\rangle|\Theta_-\rangle \qquad (1)$$

Developing this entangled resource is also important in the context of purely optical quantum communications. Indeed, the two encodings have complementary advantages[14]. Continuous variables can benefit from unconditional operations, high detection efficiencies, unambiguous state discrimination, and more practical interfacing with conventional information technology. However, they suffer from strong sensitivity to losses and intrinsically limited fidelities. On the other hand, DV approaches can achieve fidelity close to unity, but usually at the expense of probabilistic implementations. Combining the two in hybrid architectures[15,16] may offer significant advantages[17,18], particularly in the context of quantum repeaters[19–21].

Here we address the challenge of conversion between the CV and dual-rail single-photon qubits by preparing a two-mode resource state Eq. (1) and showing basic applications of it, such as remote state preparation, teleportation and entanglement swapping between the two encodings.

## Results

**Concept.** A simpler version of state Eq. (1) can be produced as sketched in Fig. 1a. We start with a weakly squeezed vacuum state with the squeezing parameter $\zeta = 0.18$, generated in a horizontally polarized mode via degenerate parametric down-conversion. This state is an excellent approximation to the positive Schrödinger's cat state $|\Theta_+\rangle$ with the amplitude $\gamma_+ = \sqrt{\zeta}$[6,12]. Generally, this approximation is only valid for cats of relatively low amplitudes, but our scheme can be used equally well with cats generated using methods that enable higher amplitudes[7,8]. The cat state then passes through a half-wave plate and polarizing beam splitter (PBS), which in combination act as a variable-reflectivity beam splitter. In the case of low reflectivity $r \ll 1$, the resulting state can be written as

$$\begin{aligned}|\psi\rangle_{\mathrm{VA,HC}} &= |0\rangle_{\mathrm{VA}}|\Theta_+\rangle_{\mathrm{HC}} + \sqrt{r}\hat{a}^\dagger|0\rangle_{\mathrm{VA}}\hat{a}|\Theta_+\rangle_{\mathrm{HC}} \\ &= |0\rangle_{\mathrm{VA}}|\Theta_+\rangle_{\mathrm{HC}} + \sqrt{r}|1\rangle_{\mathrm{VA}}\frac{\mathcal{N}_+}{\mathcal{N}_-}\gamma_+|\Theta_-\rangle_{\mathrm{HC}},\end{aligned} \qquad (2)$$

because applying a photon annihilation operator to the state $|\Theta_+\rangle$[6] transforms it into $|\Theta_-\rangle$. In the above equation, VA and HC denote, respectively, the vertical component of spatial mode A and the horizontal component of mode C.

Now let us suppose a weak horizontally polarized coherent state $|\alpha\rangle \approx |0\rangle + \alpha|1\rangle$ is injected into the input mode A of the PBS. We then obtain the state

$$\begin{aligned}|\Omega\rangle_{\mathrm{AC}} = |\alpha\rangle_{\mathrm{HA}}|\psi\rangle_{\mathrm{VA,HC}} &\approx |0\rangle_{\mathrm{HA}}|0\rangle_{\mathrm{VA}}|\Theta_+\rangle_{\mathrm{HC}} \\ &+ \alpha|1\rangle_{\mathrm{HA}}|0\rangle_{\mathrm{VA}}|\Theta_+\rangle_{\mathrm{HC}} + \beta|0\rangle_{\mathrm{HA}}|1\rangle_{\mathrm{VA}}|\Theta_-\rangle_{\mathrm{HC}},\end{aligned} \qquad (3)$$

where $\beta = \sqrt{r}\gamma_+\mathcal{N}_+/\mathcal{N}_-$ and we approximated to the first order in $\beta$ and $\alpha$. The last line in the above state corresponds to a single photon present in spatial mode A. It comprises the desired resource Eq. (1) in modes A and HC (we use $|V\rangle_{\mathrm{A}} \equiv |0\rangle_{\mathrm{HA}}|1\rangle_{\mathrm{VA}}$ and $|H\rangle_{\mathrm{A}} \equiv |1\rangle_{\mathrm{HA}}|0\rangle_{\mathrm{VA}}$ to switch between the first and second quantization notations).

In spite of a strong vacuum component in mode A, state Eq. (3) can be utilized for quantum communication protocols. Here we demonstrate the application of this state for remote state preparation[22] and teleportation[23] of a qubit from the polarization onto the CV encoding. Moreover, we show how to utilize entanglement swapping to purge the vacuum component from that state.

For remote state preparation, we project mode A onto a superposition $a|H\rangle + b|V\rangle$ by means of a polarization analyzer and a single-photon detector (SPCM) [Fig. 1b]. A click of the SPCM heralds the preparation of a CV qubit

$$(a^*\langle H| + b^*\langle V|)_{\mathrm{A}}|\Omega\rangle_{\mathrm{AC}} = a^*\alpha|\Theta_+\rangle_{\mathrm{C}} + b^*\beta|\Theta_-\rangle_{\mathrm{C}} \qquad (4)$$

(we use letter C to denote mode HC from now on, because mode VC is in the vacuum state and does not become involved in the analysis). The performance of the procedure is tested by homodyne tomography in mode C [Fig. 1c]. The values of $a$ and $b$ can be varied arbitrarily by changing the angles of half- and quarter-wave plates in the polarization analyzer; thereby CV qubits of arbitrary values can be conditionally prepared[24].

For teleportation, "Alice" prepares a heralded source photon in a polarization state $|\chi\rangle = a|H\rangle + b|V\rangle$ in an additional spatial mode B [Fig. 1d]. We then apply the Bell state projector $\langle\Psi^-| = (\langle H|\langle V| - \langle V|\langle H|)/\sqrt{2}$ to modes A and B as shown in Fig. 1e.

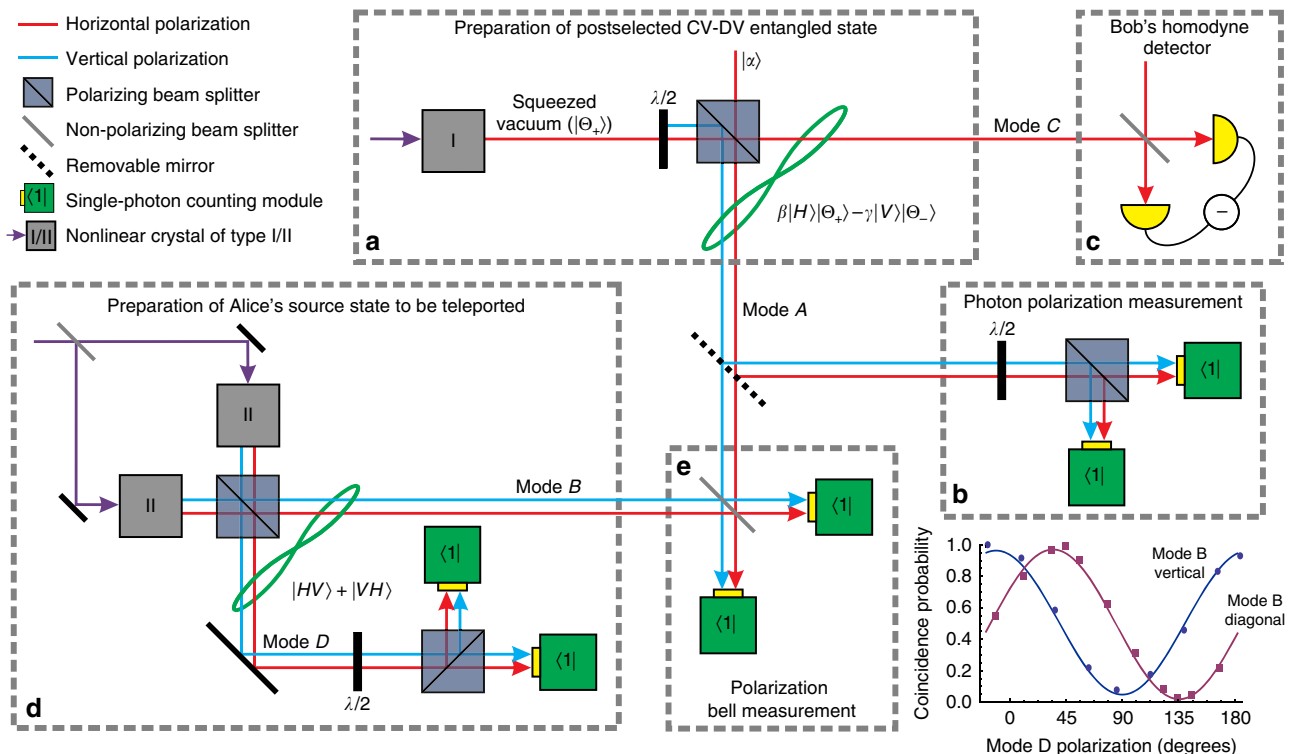

**Fig. 1** Conceptual scheme of the experiment. **a** State Eq. (3) is prepared in modes A and C. This state is equivalent to the discrete-continuous entangled state Eq. (1) conditioned on the presence of a photon in mode A. If the photon in mode A comes from $|\Theta_+\rangle$ (i.e. is vertically polarized), the state in mode C becomes $|\Theta_-\rangle$. If the photon comes from the horizontally polarized state $|\alpha\rangle$, the state in mode C remains $|\Theta_+\rangle$. The entanglement is verified by measuring the polarization of the photon in the discrete mode (**b**) and performing homodyne tomography (**c**) of the state in the continuous mode. **d** Preparation of the heralded photon in mode B whose polarization is used as the source state for teleportation. **e** Polarization Bell measurement teleports that state onto mode C. The mirror leading to part **b** is removed for the teleportation experiment. The inset shows the coincidence rate for simultaneous polarization measurements in modes B and D as a function of the polarization projection angle in mode D while a polarizer is set in mode B to project it onto either horizontal or diagonal polarizations

We obtain

$$|\phi\rangle_C = \langle\Psi^-|_{AB}\big(|\Omega\rangle_{AC}|\chi\rangle_B\big)$$
$$= \tfrac{1}{\sqrt{2}}\big(a\beta|\Theta_-\rangle_C - b\alpha|\Theta_+\rangle_C\big). \qquad (5)$$

Because the Bell measurement requires two photons to be present in its input, it will cut off the vacuum term in Eq. (3), so the state of the input photon is teleported onto "Bob's" freely propagating CV qubit in mode C.

**Remote state preparation**. We prepare state Eq. (3) as described above (see Methods for further detail), with the ratio $\beta/\alpha$ of about 0.6. In order to create a maximally entangled DV-CV state, this ratio should have been unity. However, choosing a lower value helps increasing the data acquisition rate (which was a critical parameter in this experiment) while still allowing us to see the effects we wish to observe.

We project mode A onto elements of the canonical, diagonal and circular polarization bases and perform homodyne tomography on the resulting states in mode C. A total of 2500 quadrature samples are recorded for each state. The states are then reconstructed via a maximum-likelihood algorithm[25] in the Fock basis with the reconstruction space including states up to three photons, with a correction for the homodyne detection efficiency of 0.55 [Fig. 2a]. This algorithm ensures that the reconstructed density matrices are normalized and nonnegative definite[25].

Projections onto the horizontal and vertical polarization states yield $|\Theta_+\rangle$ and $|\Theta_-\rangle$, respectively, which resemble the squeezed vacuum and photon-subtracted squeezed vacuum states[6]. Our results show fidelities of 0.99 and 0.95 with the ideal cat states $|\Theta_+\rangle$ and $|\Theta_-\rangle$ of amplitudes $\gamma_+ = 0.45$ and $\gamma_- = 0.90$, respectively, where the fidelity between states $\hat{\rho}_1$ and $\hat{\rho}_2$ is defined as $F = \left(\text{Tr}\sqrt{\hat{\rho}_1^{\frac{1}{2}}\hat{\rho}_2\hat{\rho}_1^{\frac{1}{2}}}\right)^2$. A difference between $\gamma_+$ and $\gamma_-$ is inherent in the preparation method; theoretically, we expect $\gamma_-/\gamma_+ = \sqrt{3}$[12]. Note that the Wigner function of the experimentally reconstructed state $|\Theta_-\rangle$ exhibits negativity even without efficiency correction.

Projecting mode A onto superpositions of $|H\rangle$ and $|V\rangle$ produces analogous CV superpositions Eq. (4). In particular, projecting onto the diagonal basis yields states that approximate coherent states, with the Wigner functions exhibiting a characteristic shape of a displaced Gaussian peak. Deviation of the Wigner functions from the Gaussian shape for these superpositions is mainly due to the different amplitudes $\gamma_+ \neq \gamma_-$ of the constituent cats, as well as experimental nonidealities. The coherent nature of these superpositions is also evidenced by their density matrices in the subspace spanned by the basis $\{|\Theta_+\rangle, |\Theta_-\rangle\}$ shown in the bottom row of Fig. 2a. The fidelity with the theoretically expected states Eq. (4) exceeds 0.93 in all cases.

The states displayed in Fig. 2a can be used to fully reconstruct the component of the DV–CV state in modes A and C, projected

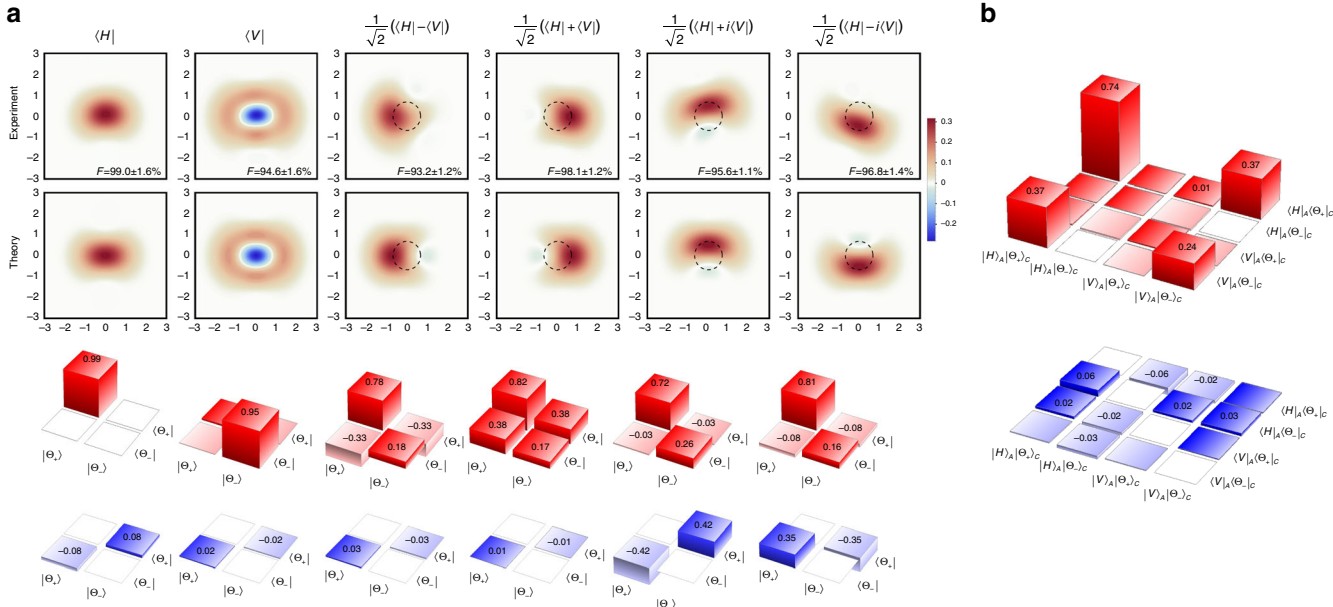

**Fig. 2** Results of the remote state preparation experiment. **a** The state of the CV mode C when the DV mode A is projected onto various polarization states. Top rows: experimental and corresponding theoretical Wigner functions. Red color indicates positive values, blue—negative. Theoretical Wigner functions correspond to superpositions Eq. (5) of cat states. The dashed circles correspond to the half-height of the vacuum state Wigner function. Superpositions of cat states that correspond to the projections onto diagonal polarizations approximate coherent states, hence their Wigner functions exhibit displacement relative to the origin of a phase space. The fidelities between the experimentally acquired states and theoretically expected superpositions of cat states are shown. Bottom rows: projections of the experimental density matrices onto the subspace spanned by $\{|\Theta_+\rangle, |\Theta_-\rangle\}$. The real parts are shown in red, imaginary in blue. Note that these density matrices are not normalized because of the imperfect overlap of the reconstructed states with this subspace. **b** Density matrix of the reconstructed DV-CV state in modes A and C

onto the subspace corresponding to a single photon in mode A (see Methods). The density matrix of this state [Fig. 2c] has a fidelity of $93.1 \pm 2.4\%$ with the state Eq. (1). The fidelity with the maximally entangled state, given by Eq. (1) with $\alpha = \beta$, equals $84.4 \pm 2.2\%$. This evidences entanglement of this state, because the fidelity of a separable state with a maximally entangled biqubit state cannot exceed $1/2$[26].

**Teleportation.** To produce a source photon, we first prepare a photon pair in a polarization entangled Bell state $|\Psi^+\rangle_{\mathrm{BD}}$. This is realized by overlapping the outputs of parametric down-conversion from two crystals, in each of which a collinear, frequency-degenerate pair of the form $|HV\rangle$ is produced, on a PBS [see Fig. 1d, ref.[27] and Methods]. By measuring the polarization of the photon in mode D, we prepare a heralded photon in mode B in a certain polarization state. The performance of the method is illustrated by the inset in Fig. 1.

Subsequently, we perform a Bell measurement on modes A and B using the technique from the original experiment on quantum teleportation of a photon polarization state[28]. Namely, these modes are subjected to interference on a symmetric non-PBS. If a single photon is present in each beam splitter input, the Hong-Ou-Mandel effect[29] forces the photons to emerge in the same output spatial mode unless the input is in the Bell state $|\Psi^-\rangle_{\mathrm{AB}}$ [Fig. 1e]. Thus a coincidence detection event in both beam splitter outputs projects its input onto that state. In this way, we can detect one of the four Bell states. While there exists a linear optial protocol that enables increasing this number to two[30], all four Bell states cannot be distinguished by means of linear optics only[31,32]. This problem is common to all schemes that involve teleportation of polarization states of single photons.

This method also suffers from an issue raised in Braunstein and Kimble's correspondence[33] on ref.[28]. The coincidence event can occur not only due to one photon coming from each of the modes A and B, but also when both photons come from the same mode. The latter events, which we refer to as "double A/B events" result in false positive Bell state detection. In our case, only the double B events are of concern because the photon in mode B is heralded and $\alpha, \beta \ll 1$. The probability $p_{\mathrm{dB}}$ of these events can be reduced by lowering the pumping of parametric down-conversion in modes B and D[34] to the level such that $p_{\mathrm{dB}} \ll p_{\mathrm{good}}$, where $p_{\mathrm{good}}$ is the probability of a true positive Bell detection event in which the two photons come from different modes. In the actual experiment, we have $p_{\mathrm{good}}/p_{\mathrm{dB}} = 1$–$3$ (see Methods). The variation of this ratio is due to the dependence of the "good" Bell detection probability on the input state: $p_{\mathrm{good}} \propto \||\varphi\rangle\|^2 = (|a\beta|^2 + |b\alpha|^2)/2$, where the state $|\varphi\rangle$ is given by Eq. (5).

A teleportation event is heralded by a triple coincidence photon detection in modes A, B (Bell detection) and one of the photons in mode D (input photon heralding). The rate of these events is about 0.015 Hz. Because of this extremely low rate, we limit the input polarization states chosen for the test of the protocol to four: $|H\rangle$, $|V\rangle$ and $(|H\rangle \pm |V\rangle)/\sqrt{2}$. For each of these states, 1500 homodyne measurement of the state in the output CV mode C are collected and the state reconstruction is implemented with compensation for the homodyne detection efficiency of 0.55.

Figure 3a shows the results of this reconstruction. The primary detrimental effect on the teleportation fidelity is false positive Bell state detections due to double B events. In such an event, no photon annihilation in mode C takes place, resulting in that mode containing the state $|\Theta_+\rangle$. As a result, the teleportation output

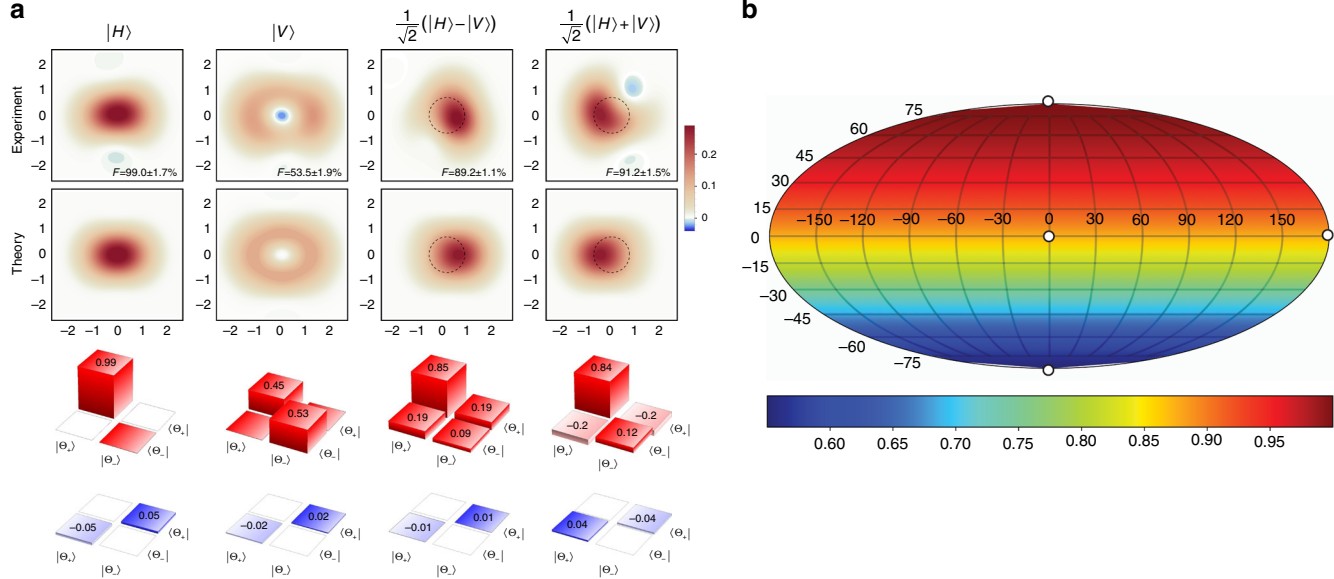

**Fig. 3** Results of the teleportation from a polarization qubit in mode B onto a CV qubit in mode C. **a** Wigner functions and density matrices of the teleported states for four input polarization states [same notation as in Fig. 2, the fidelities are calculated with respect to the theoretically expected superpositions Eq. (5) of cat states]. Theoretical Wigner functions are calculated according to Eq. (6). **b** Teleportation fidelity based on the theoretical model Eq. (6) which agrees well with the experimental results in **a**, calculated for the entire Bloch sphere. The white dots represent the input states of the teleportation experiment

can be written as

$$\hat{\rho}_{out} = \frac{p_{good}}{p_{good} + p_{dB}} \mathcal{N}(|\varphi\rangle\langle\varphi|) + \frac{p_{dB}}{p_{good} + p_{dB}} |\Theta_+\rangle\langle\Theta_+|, \quad (6)$$

where $\mathcal{N}$ denotes normalization. The ratio $p_{dB}/p_{good}$ reaches a maximum value of ~1 for the input state $|\chi\rangle = |H\rangle$ (so $a = 1$ and $b = 0$), in which case the fidelity of the output state approaches 53% [Fig. 3a, second column]. For input states with a significant fraction of $|V\rangle$, the fidelity is much higher.

The simple model of Eq. (6) describes the four teleported states with a ≥93% fidelity. We use the above model to determine the teleportation fidelity for an arbitrary polarization qubit as input [Fig. 3b]. We find the mean fidelity over the full Bloch sphere to equal 80%, which is above the classical benchmark of 2/3[35].

**Entanglement swapping**. An important alternative interpretation of our experiment is entanglement swapping[36]. If one considers the Bell measurement in modes A and B without accounting for the measurement in mode D, one obtains

$$\langle\Psi^-|_{AB}\left(|\Omega\rangle_{AC}|\Psi^+\rangle_{BD}\right) = \alpha|H\rangle_D|\Theta_+\rangle_C - \beta|V\rangle_D|\Theta_-\rangle_C. \quad (7)$$

The Bell measurement cuts off the first term in Eq. (3), thereby heralding a freely propagating resource state Eq. (1) in modes C and D.

A proper realization of this scheme would require a postselection-free source of photon pairs in a Bell state. Such sources are accessible with existing experimental methods, capable of producing photons in the heralded[37–39] fashion; on-demand source are also emerging[40,41]. However, constructing a heralded entangled pair source in our experiment would result in a prohibitively low state production rate. A "conventional" Bell state source with a dominating vacuum component, such as ours, will result in a large fraction of false positive double A or double B Bell state detection events. This will preclude the production of state Eq. (7) with high fidelity.

However, the results in Fig. 3a can be interpreted to characterize the entanglement swapping output state in the a

posteriori manner: conditioned on a photon detection event in mode D. This event guarantees that a photon has been present in mode B and thereby eliminates the double A events from the analysis.

The four reconstructed states can be used to determine a lower bound of the fidelity between the entanglement swapping output and the maximally entangled state[42] (see Methods). We find this lower bound to be $0.56 \pm 0.03$, which is above the entanglement threshold of $1/2$[26].

## Discussion

To sum up, we realized a method for hybrid resource generation between a CV qubit and a qubit encoded in the polarization of a single photon. The present realization suffers from several issues that prevent its direct use in quantum information processing. First, the amplitude of the cat state is below the values $|\alpha| \gtrsim 1$ that ensure sufficient orthogonality of the component coherent states and are thus optimal for practical applications[4,5]. Second, the successful event rate is very low. Third, false positive Bell detection events degrade the teleportation fidelity and prevent production of freely propagating hybrid entangled states.

However, these issues can be rectified using existing or emerging technology. First, the use of non-postselected entangled photon pair sources will eliminate the false-positive events. If these sources are of on-demand nature, the issue of low count rates is also mitigated. Finally, using input cat states that are more sophisticated than simple squeezed vacuum allows raising the amplitudes of these states. For example, squeezed or unsqueezed single photons from on-demand sources will permit producing cats of high amplitudes without significantly compromising the count rates[8,43].

As a concluding remark, the technique reported here can be useful for the interconversion not only between polarization and continuous-variable qubits, but also between single-rail and dual-rail photonic qubits. This is because, in the limit of low squeezing, the state $|\Theta_+\rangle$ is close to the vacuum state while the state $|\Theta_-\rangle$ is well-approximated by the single photon. However, the specific scheme of our experiment may not be optimal for this application

because the production rate of the corresponding entangled states would tend to zero in this limit. In the future, we plan to study possible modifications of the scheme that would result in higher productivities.

## Methods

**Experimental setup.** The initial 1.5-dB single-mode squeezed vacuum state in mode C is produced by degenerate parametric down-conversion in a periodically poled potassium titanyl phosphate crystal (PPKTP, Raicol) under type-I phase-matching conditions. The crystal is pumped with ~20 mW frequency-doubled radiation of the master laser (Ti:Sapphire Coherent Mira 900D, with a wavelength of 780 nm, repetition rate of $R = 76$ MHz and pulse width of 1.5 ps)[44].

For the preparation of polarization-entangled photon pairs [Fig. 1d], a polarization interferometer scheme is used[27]. A symmetric beam splitter at the entrance of the interferometer splits the ~5-mW pump beam into two equal parts that are directed into PPKTP crystals in each path. Parametric down conversion in each crystal occurs in a collinear, frequency-degenerate, type-II regime and generates a two-mode squeezed vacuum state $|0\rangle_H|0\rangle_V + \lambda|1\rangle_H|1\rangle_V + \lambda^2|2\rangle_H|2\rangle_V + O(\lambda^3)$. After a polarizing beamsplitter at the end of interferometer, the orthogonally polarized modes from the two arms become temporarily and spatially indistinguishable in each of the two output modes B and D.

The path length difference between the two arms of the interferometer is locked. The feedback for the lock is obtained from the interferometric signal of two pump beams and is applied to a piezoelectric transducer in one of the arms. The resulting polarization-entangled state is characterized by simultaneous polarization analysis in modes B and D in linear bases. The coincidence count rate is measured with different angles of the half-wave plate in mode D while a polarizer inserted in front of the detector in mode B is kept constant. This rate exhibits a characteristic sinusoidal shape with a visibility of 97% (Fig. 1, inset).

Photon detection is implemented by fiber-coupled SPCMs (Excelitas). In mode D, two SPCMs are used to detect orthogonal polarization states, which permits simultaneous acquisition of the quadrature data corresponding to the teleportation of both these states. The data acquisition is triggered by a home-made delay/coincidence circuit based on an Artix-7 35T field-programmable gate array.

The relative phase between the two terms of the CV–DV entangled state is determined by the phases of the input coherent and squeezed states, whose difference must therefore be kept constant. We measure both these phases with respect to that of the local oscillator used for homodyne detection. The phase of the squeezed vacuum is determined from the quadrature variance acquired by the homodyne detector in HC without conditioning on single-photon detection events. To measure the phase of the coherent state, we prepare it with a significant vertical polarization component. This component is then reflected into the mode VC and measured with an auxilliary homodyne detector. The difference between the two phases is locked to zero by means of a feedback signal applied to a piezoelectric transducer in the path of the input coherent state. the phase of the local oscillator, on the other hand, is varied during the experiment.

Because we keep track of the evolution of the HC mode phase in time, we also know its phase at each moment the event of interest occurs. For each such event, the phase and quadrature value of the HC mode are recorded. Because these events are relatively rare, the phases in the acquired data set are randomly distributed from 0 to $2\pi$. This data set is fed directly to the reconstruction algorithm[25]; no binning of the phases or quadratures is implemented. In this way, the criterion of informational competeness for homodyne tomography[45] is satisfied.

In order to evaluate the accuracy of reconstruction method with respect to statistical errors, we use bootstrapping. We randomly generate simulated quadrature data sets that correspond to the reconstructed state and apply the maximum-likelihood algorithm to these data sets, thereby obtaining a set of secondary density matrices that approximate the original reconstructed one. The fidelity uncertainties quoted throughout the paper are determined from this secondary set according to $\Delta F = \sqrt{\langle\Delta F'^2\rangle + \langle F - F'\rangle^2}$. Here $\langle\Delta F'^2\rangle$ is the variance of the fidelities of the secondary set with respect to the theoretically expected state and $\langle F - F'\rangle$ is the systematic bias that the fidelity of the secondary set exhibits with respect to the original reconstructed state[46]. In all cases, the contribution of this bias to $\Delta F$ did not exceed about one-tenth of $\sqrt{\langle\Delta F'^2\rangle}$.

The total quantum efficiency of homodyne detection, 55%, is determined from the analysis of the negative cat state $|\Theta_-\rangle$ generated in mode C conditioned on a photon detection in VA. The main efficiency reduction factors are optical losses (90% cumulative transmissivity of all optical elements, in addition to the tapping beamsplitter which also has a $1 - r = 90\%$ transmissivity), mode matching between the signal and local oscillator (81%) and the quantum efficiency of the homodyne detector (86%)[47,48].

**Photon count rates.** With mode A blocked, the polarization-entangled pairs generated in the two type-II crystals produce count rates of $R_{B,D} \sim 4 \times 10^3\,\text{s}^{-1}$ in each of the SPCMs in mode D and the Bell detector. The coincidence rate between each pair of SPCMs in mode D and the Bell detector is ~20 s$^{-1}$, meaning that the single-photon detection efficiency is $\eta_{SPCM} = 0.01$. Such a low efficiency is

explained by the presence of narrowband (0.2 nm) filters in front of each SPCM[44], in addition to the usual linear losses. Based on these numbers, the probability of an undesired double-B event, coincident with a click in and one of the detectors in mode D, can be estimated as $p_{dB} = \frac{3}{2}\eta_{SPCM}R_B^2/R^2 \approx 4 \times 10^{-11}$.

When mode B is blocked, the count rate in each of the two SPCMs in the Bell detector is $R_\alpha = 18 \times 10^3\,\text{s}^{-1}$ due to the coherent state and $R_\beta = 6 \times 10^3\,\text{s}^{-1}$ due to the squeezed vacuum state, so $R_\beta/R_\alpha = \beta^2/\alpha^2 = 1/3$. The probability of a "good" triple coincidence event of the two SPCMs in the Bell detector and one of the detectors in mode D is therefore estimated as $p_{good} = \eta_{SPCM}R_B\left(b^2 R_\alpha + a^2 R_\beta\right)/R^2 \approx 4\text{–}12 \times 10^{-11}$. The expected total triple coincidence event rate, $R(p_{good} + p_{dB}) = 6\text{–}12 \times 10^{-3}\,\text{s}^{-1}$, is consistent with the rate observed in the experiment within a factor of one and a half.

**Two-mode state reconstruction.** The DV-CV state $\hat\rho$ in modes A and C [Fig. 2b] can be recovered from the six density matrices in mode C [Fig. 2a] that represent the projections $\langle\pi|\hat\rho|\pi\rangle$ of that state onto various polarization states $|\pi\rangle$ in mode A. To find $\hat\rho$, we write it in a generic form

$$\hat\rho = |H\rangle\langle H| \otimes \hat\rho_{HH} + |V\rangle\langle V| \otimes \hat\rho_{VV} + |H\rangle\langle V| \otimes \hat\rho_{HV} + |V\rangle\langle H| \otimes \hat\rho_{HV}^\dagger \quad (8)$$

where, e.g., $\hat\rho_{HV} = \langle H|\hat\rho|V\rangle$. The first two terms in the above expression are obtained directly from the first two columns in Fig. 2a. The remaining two terms are evaluated from the remaining four columns according to

$$\hat\rho_{HV} = \frac{1}{2}(\langle D|\hat\rho|D\rangle - \langle A|\hat\rho|A\rangle + i\langle L|\hat\rho|L\rangle - i\langle R|\hat\rho|R\rangle), \quad (9)$$

where $|A,D\rangle = \frac{|H\rangle \pm |V\rangle}{\sqrt{2}}$ and $|R,L\rangle = \frac{|H\rangle \pm i|V\rangle}{\sqrt{2}}$ are the diagonal and circular polarization states.

**Entanglement criterion.** In contrast to the state of modes A and C analyzed above, the CV-DV state of modes D and C, obtained after entanglement swapping, cannot be reconstructed because only its projections onto the canonical and diagonal polarization states are known [Fig. 3a]. However, these data can be used to estimate the lower bound of the fidelity with the maximally entangled state $|\Psi_{ME}\rangle = \frac{1}{\sqrt{2}}(|H\rangle|\Theta_+\rangle - |V\rangle|\Theta_-\rangle)$. To this end, we follow the argument of ref.[42] and write

$$F = \langle\Psi_{ME}|\hat\rho|\Psi_{ME}\rangle$$
$$= \frac{1}{2}\left(\hat\rho_{H\Theta_+,H\Theta_+} + \hat\rho_{V\Theta_-,V\Theta_-} - \hat\rho_{H\Theta_+,V\Theta_-} - \hat\rho_{V\Theta_-,H\Theta_+}\right) \quad (10)$$

where $\hat\rho$ is the density matrix of the DV-CV state in question written in the basis $\{|H\rangle, |V\rangle\} \otimes \{|\Theta_+\rangle, |\Theta_-\rangle\}$. The first two terms in Eq. (10) are obtained from the first two columns in Fig. 3a taking into account the probabilities of occurrence of the corresponding polarization states in mode D. The sum of the last two terms can be estimated as follows:

$$\hat\rho_{H\Theta_+,V\Theta_-} + \hat\rho_{V\Theta_-,H\Theta_+} = \hat\rho_{D\Theta_D,D\Theta_D} + \hat\rho_{A\Theta_A,A\Theta_A} - \hat\rho_{D\Theta_A,D\Theta_A}$$
$$- \hat\rho_{A\Theta_D,A\Theta_D} - \left(\hat\rho_{V\Theta_+,H\Theta_-} + \hat\rho_{H\Theta_-,V\Theta_+}\right) \quad (11)$$

where $|\Theta_{A,D}\rangle = \frac{1}{\sqrt{2}}(|\Theta_+\rangle \pm |\Theta_-\rangle)$. The first line in the right-hand side of the above equation is obtained from the last two columns in Fig. 3a. The last line can be bounded by

$$\left|\hat\rho_{V\Theta_+,H\Theta_-} + \hat\rho_{H\Theta_-,V\Theta_+}\right| \leq 2\sqrt{\hat\rho_{H\Theta_-,H\Theta_-}\hat\rho_{V\Theta_+,V\Theta_+}}. \quad (12)$$

Combining Eqs. (10–12) yields a bound on the fidelity $F$. Since the lower bound exceeds $1/2$, the state is entangled[26]. We note that, while fidelity has been criticized as a general criterion for state similarity[49], the present fidelity-based performance criterion[26], as well as the criterion of ref.[35], are universal for any qubit or e-bit regardless of its physical nature, and therefore can be applied in the context of the present work.

## Data availability
The data that support the findings of this study are available from the corresponding author upon request.

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

## Acknowledgements

We thank Marco Barbieri for inspiring discussions. A.L.'s research is supported by NSERC and CIFAR.

## Author contributions

The project was conceived and planned by D.S., A.U., A.P., and A.L.. The experiment was performed by D.S., A.U., A.P., E.T., V.N., and A.K. The data were analysed by D.S., A.U., E.T., A.K., and A.L. The paper was written by D.S., E.T. and A.L.

## Additional information

**Competing interests:** The authors declare no competing interests.

