## [Peer Review File · Nature Communications]

Reviewers' comments:

Reviewer #1 (Remarks to the Author):

The work by Sychev and co-workers reports about the experimental demonstration of well-known basic quantum information protocols such as quantum teleportation, state preparation and entanglement swapping, utilizing hybrid entanglement as resource. As already demonstrated, hybrid entanglement provides a connection between qubits encoded with discrete variables (DV), such as polarization state of a single photon, and qubits encoded with continuous variables (CV), such as amplitude and phase of a coherent state. In this work, a clever scheme generates hybrid entanglement compatible with the basic experimental setups to achieve teleportation and entanglement swapping. In particular, DV qubit is encoded using the dual-rail configuration where a single photon can occupy one of two orthogonal spatial modes; while the CV qubit is encoded using Schrodinger cat states obtained from squeezed light.

The manuscript is very clear and the results are very well presented and convincing. Since the work represents an important step toward hybrid quantum information processing, I highly recommend this manuscript for publication.

However, I have some classifications to ask about the reconstruction method used to retrieve the results of Fig. 2 and 3. It would be interesting to know which kind of basis has been used in the reconstruction algorithm, number basis or Schrodinger cat basis? If usual number basis is used, how large is the reconstructed Hilbert space? In addition, I would like to ask how many phase bins the authors used for the reconstruction, and if the local oscillator phase of Bob's homodyne detector was locked to a set of values or slowly swept during quadrature acquisition.

Reviewer #2 (Remarks to the Author):

This paper describes an experimental demonstration of various notions and protocols in optical quantum information processing involving both discrete polarization-encoded qubits and continuous (phase-encoded) "cat-state" qubits. An immediate application of such schemes would be the conversion between different encodings, each adapted to a corresponding, different degree of freedom of a distinct physical platform for quantum interfaces. While similar demonstrations have been performed already involving "single-rail" instead of "dual-rail" (of which polarization encoding is a special case) qubits, a dual-rail implementation is, both from a conceptual and from a practical point of view, more important. This is why I believe that the present work is significant.

Overall the paper is very well written, the topic is well motivated, and the state of the art well described. Also the detailed explanations concerning the actual experiment are of high readability. This, together with my initial assessment that the topic is important, makes me conclude that I am happy to recommend the paper for publication in Nature Communications.

Nonetheless, there are a few points that the authors should address before publication:

1.) While the theoretical concept and the experimental realization are well described on their own, I feel that the link between the two still lacks a certain degree of clarity. More specifically and in particular, the experimental results as illustrated by Fig.2a) are a bit unclear and unsatisfactory to me. The top row shows the Wigner functions of the projected CV states conditioned upon the different measurement results for the DV part. All six states have fairly high fidelities. In the text it is said that the reference states are "the ideal cat states", but also a distinction is mentioned between the theoretically expected states and those based on the preparation method of the experiment. Could the authors possibly be more specific on what is meant by "the ideal states"?

Related with this, in case of the H- and V-projections, one would expect the Θ_+ and Θ_- states for the CV part, as explained in the text. The former is a Gaussian state, whereas the latter is non-Gaussian. Is this / shouldn't this be visible to some extent in the Wigner contours of the figure? Moreover, only the latter state exhibits Wigner negativity. This is kind of mentioned in the text, but the figure is not so compelling in this context. Could one illustrate this aspect via 3D plots of the Wigner functions? Some of the other projections would ideally, considering perfect cat states, lead to simply coherent states of different phases for the conditional states; so one would expect "simple" Gaussian Wigner functions (indicated by the dashed black circles in the figure?), whereas the experimentally determined Wigner functions in those cases look rather non-Gaussian. This is certainly related to the fact that α and β are unequal and it may also be related to the approximate nature of the cat states. Ideally, one would expect more non-Gaussian features for the H- and V-projections compared to the (H \pm V)-projections. Maybe these aspects could be made clearer, comparing theoretical expectations with the present experimental results.

2.) In the above-mentioned discussion on page 3, right column, maybe the conditionally obtained Θ_- state should be referred to as "photon-subtracted squeezed vacuum state" in order to connect that discussion to the previous one below the equation without number (the equation between Eqs.(1) and (2)); the additional, equivalent characterisation as "squeezed single-photon state" may remain there as well.

3.) One could argue that the results for the teleportation experiment are a bit incomplete for two reasons. First, the input states to the teleporter are drawn only from a set of four states (formally these are the four qubit Pauli-Z and X eigenstates). A "more complete" sample would be one that also includes the Pauli-Y eigenstates, like in the remote state preparation experiment (is/was it experimentally easier to project onto the six eigenstates compared to preparing the corresponding six source photon states?). Second, the teleportation is conditioned upon that Bell measurement result that, on Bob's side, does not require any additional operations by Bob; this happens only in one quarter of all cases. The first point is a complication encountered in most teleportation experiments: what is a good representative sample of input states and how to create all those states efficiently? In my opinion, the whole package of experimental demonstrations including remote state preparation, teleportation, and entanglement swapping (though only "a posteriori") is sufficiently compelling. I also appreciate that the authors discuss in detail the initial debate concerning the first Zeilinger teleportation experiments (Refs.27 and 29) and its subtleties, which were eventually circumvented in a subsequent experiment (Ref.30). The present experiment basically makes use of the "trick" of Ref.30.

Reviewer #3 (Remarks to the Author):

The authors generate and characterize a hybrid entanglement between a discrete-variable (DV) qubit encoded in polarization states and a continuous-variable (CV) qubit encoded in coherent superposition states. Such a hybrid entangled state constitutes a crucial optical resource in the so-called heterogeneous quantum network that combines DV and CV techniques for optical quantum information processing. In their paper, several basic applications of the hybrid entanglement are experimentally implemented, such as remote state preparation, teleportation and entanglement swapping between the two encodings. In general, the paper is well organized, clearly presented and easy to follow, albeit with some typos and ambiguities in the text. However, from my perspective the given experimental results have not yet reached the significant and firm level to justify publication in Nature Communications. The reasons are elaborated below.

1. As recognized by the authors in the introductory part, in continuous-variable quantum information processing, the amplitude of the coherent state should be sufficiently large to ensure the orthogonality for computational bases. However, the small amplitude of about 0.5 given in this paper results in significant overlap between two coherent state bases, which even precludes the

implementation of fault-tolerant operations (requiring coherent states with a moderate amplitude $|\alpha| \approx 1$). In this sense, the demonstrated state is not ready to use in subsequent protocols in hybrid CV-DV quantum information processing. Additionally, the entangled state is generated in a posteriori manner, and thus cannot propagate freely as a traveling field, which prohibits its applications in the context of quantum repeaters and quantum networks.

2. The success rate is intrinsically low in the reported experimental schemes, which not only limits the practical usage, but also renders it impossible to implement fully tomography of the generated two-mode states. Due to the limited number of quadrature values (about 2500 for each state), the authors have to conduct the quantum state reconstruction in the subspace. As a result, the density matrices shown in Fig. 2 are not normalized. As far as I know, it is important to acquire sufficiently large amount of quadrature samples for quantum state reconstruction to reduce numerical errors in the maximum-likelihood algorithm. In the experiment, the correction of close to 1/2 detection efficiency in the maximum-likelihood algorithm may lead to a large numerical uncertainty for the reconstructed states, especially in the case of small amount of quadrature samples. Rigorously, the fidelities obtained in the paper should be given with corresponding uncertainties, particularly when they are compared with classical bounds.

3. The fidelities presented in the text are corrected for a detection loss as large as 45%. I suppose that the raw fidelities without correction are not high enough to surpass the classical benchmark in the demonstration of entanglement and teleportation. To be rigorous, the classical bound for quantum witness should be calculated taking account of the detection efficiency. This is particularly important for continuous-variable non-classical states since they are notoriously fragile to losses. For continuous-variable states, fidelity is a misleading figure of merit in some cases. For instance, it is commonly known that a state with a smaller energy is more robust to losses, thus easier to maintain a high fidelity with the original state. The classical bounds in this paper should depend on the size of coherent state as the basis in CV encoding.

In conclusion, I have no doubt that the work presented in this paper contributes valuable investigations towards the optical hybrid quantum information processing involving CV and DV sources and techniques. However, from my point of view the Achilles' heel of the obtained results is the small size of the prepared cat state. Consequently, the non-negligible overlap between two coherent states as computational bases practically precludes the so-called wave-like encoding as claimed in the title of the paper. Unfortunately, the limited preparation rate makes it hard to further use quantum amplification to enlarge the amplitude of the computational basis state. Therefore, based on the quality of the presented results, I am not persuaded to accept the paper for publication in this high-standard journal.

=====

RESPONSE TO REFEREE #1

The manuscript is very clear and the results are very well presented and convincing. Since the work represents an important step toward hybrid quantum information processing, I highly recommend this manuscript for publication.

We thank the reviewer for the positive evaluation of our work.

However, I have some classifications to ask about the reconstruction method used to retrieve the results of Fig. 2 and 3. It would be interesting to know which kind of basis has been used in the reconstruction algorithm, number basis or Schrodinger cat basis? If usual number basis is used, how large is the reconstructed Hilbert space? In addition, I would like to ask how many phase bins the authors used for the reconstruction, and if the local oscillator phase of Bob's homodyne detector was locked to a set of values or slowly swept during quadrature acquisition.

The reconstruction of the state was carried out in the photon number basis up to 3 photons. Because the number of data points was limited, a larger dimension of the reconstruction Hilbert space would result in overfitting. The phase of the local oscillator is indeed slowly changed with time. We do not bin the phases; the reconstruction algorithm is explained in detail in Ref. [25]. We added these details in the description of the experiment and in the Methods.

=====

RESPONSE TO REFEREE #2

Overall the paper is very well written, the topic is well motivated, and the state of the art well described. Also the detailed explanations concerning the actual experiment are of high readability. This, together with my initial assessment that the topic is important, makes me conclude that I am happy to recommend the paper for publication in Nature Communications.

We thank the reviewer for the positive evaluation of our work.

The top row shows the Wigner functions of the projected CV states conditioned upon the different measurement results for the DV part. All six states have fairly high fidelities. In the text it is said that the reference states are "the ideal cat states", but also a distinction is mentioned between the theoretically expected states and those based on the preparation method of the experiment. Could the authors possibly be more specific on what is meant by "the ideal states"?

In the new version of the article, we clarify that "the ideal cat states" are $|\Theta_+\rangle$ and $|\Theta_-\rangle$ defined in the Introduction.

Related with this, in case of the H- and V-projections, one would expect the Θ_+ and Θ_- states for the CV part, as explained in the text. The former is a Gaussian state, whereas the latter is non-Gaussian. Is this / shouldn't this be visible to some extent in the Wigner contours of the figure? Moreover, only the latter state exhibits Wigner negativity. This is kind of mentioned in the text, but the figure is not so compelling in this context. Could one illustrate this aspect via 3D plots of the Wigner functions?

In Fig. 2(a), negative values are represented by the blue color. The second column, corresponding to the V-projection, contains a negative dip, which demonstrates a strongly non-Gaussian state. In contrast, the H-projected plot is similar to a squeezed vacuum state, which is Gaussian. 3D images, when placed in the

same position in the article, would not be big enough to be informative, in particular, to show the coherent displacements of the diagonal projections. If there are no serious objections, then we would like to leave pictures in the article as they are. However, we added a statement in the caption clarifying the color code.

Some of the other projections would ideally, considering perfect cat states, lead to simply coherent states of different phases for the conditional states; so one would expect "simple" Gaussian Wigner functions (indicated by the dashed black circles in the figure?), whereas the experimentally determined Wigner functions in those cases look rather non-Gaussian. This is certainly related to the fact that α and β are unequal and it may also be related to the approximate nature of the cat states. Ideally, one would expect more non-Gaussian features for the H- and V-projections compared to the (H \pm V)-projections.

The deviation of the diagonal projections from the Gaussian shape are primarily due to the different amplitudes of the positive and negative cat, as well as experimental imperfections. We now write this explicitly in the section on remote state preparation. Interestingly, the effect of $a \neq b$ is small. This is because this inequality is compensated in Eq. (3) by the inequality of the normalization factors N_+ and N_- that enter when the cat states are expressed through the coherent states.

Maybe these aspects could be made clearer, comparing theoretical expectations with the present experimental results.

We have now included the full set of theoretical Wigner functions into Fig. 2(a) and 3(a), so direct comparison is possible.

In the above-mentioned discussion on page 3, right column, maybe the conditionally obtained Θ_- state should be referred to as "photon-subtracted squeezed vacuum state" in order to connect that discussion to the previous one below the equation without number (the equation between Eqs.(1) and (2)); the additional, equivalent characterisation as "squeezed single-photon state" may remain there as well.

We agree with this remark and made the appropriate change in the article.

One could argue that the results for the teleportation experiment are a bit incomplete for two reasons. First, the input states to the teleporter are drawn only from a set of four states (formally these are the four qubit Pauli-Z and X eigenstates). A "more complete" sample would be one that also includes the Pauli-Y eigenstates, like in the remote state preparation experiment (is/was it experimentally easier to project onto the six eigenstates compared to preparing the corresponding six source photon states?).

Indeed, as Referee said, one would acquire more complete information by measuring the teleportation output for a larger number of inputs. We limited this number to four due to the extremely low event rate. We added a comment to this effect in the appropriate section of the paper.

Second, the teleportation is conditioned upon that Bell measurement result that, on Bob's side, does not require any additional operations by Bob; this happens only in one quarter of all cases.

Indeed, only one out of four Bell states are detected. Adding PBSs and another pair of photon counting modules would increase this number to two. However, it was proven that all four Bell states can't be distinguished by only linear optics. We added a comment to this effect.

=====

RESPONSE TO REFEREE #3

The authors generate and characterize a hybrid entanglement between a discrete-variable (DV) qubit encoded in polarization states and a continuous-variable (CV) qubit encoded in coherent superposition states. Such a hybrid entangled state constitutes a crucial optical resource in the so-called heterogeneous quantum network that combines DV and CV techniques for optical quantum information processing. In their paper, several basic applications of the hybrid entanglement are experimentally implemented, such as remote state preparation, teleportation and entanglement swapping between the two encodings. In general, the paper is well organized, clearly presented and easy to follow, albeit with some typos and ambiguities in the text. However, from my perspective the given experimental results have not yet reached the significant and firm level to justify publication in Nature Communications. The reasons are elaborated below.

We thank the reviewer for recognizing the significance of the problem addressed in our paper. Indeed, as the reviewer writes, the “hybrid entangled state constitutes a crucial optical resource” in quantum networking. In other words, the lack of a method for producing this resource has been an important outstanding problem. The significance of our work consists in presenting a conceptual solution to this problem and demonstrating its experimental realization.

The reviewer argues that our paper should not be accepted for publication because, as (s)he writes in the subsequent paragraph, “the demonstrated state is not ready to use in subsequent protocols”. We respectfully submit that developing a conceptual solution and bringing it to the level of a plug-and-play prototype that is ready to use in practical protocols are fundamentally different tasks. It is in the nature of basic research that not all problems can be solved at once. Immediate practical applicability should not be used as a publication criterion for basic research papers. This is especially the case with our work because the issues pointed out by the reviewer can be addressed with the current level of technology, as we discuss below.

Having said that, we ensured in the revised version that the reader is aware of the existing issues and the ways to address them by adding two paragraphs in the conclusion where these matters are detailed.

As recognized by the authors in the introductory part, in continuous-variable quantum information processing, the amplitude of the coherent state should be sufficiently large to ensure the orthogonality for computational bases. However, the small amplitude of about 0.5 given in this paper results in significant overlap between two coherent state bases, which even precludes the implementation of fault-tolerant operations (requiring coherent states with a moderate amplitude $|\alpha| \approx 1$). In this sense, the demonstrated state is not ready to use in subsequent protocols in hybrid CV-DV quantum information processing.

While our scheme is demonstrated with low-amplitude cats, it can be directly applied to higher-amplitude cats produced by any of the existing protocols. We added a comment and references regarding this matter in the introductory part and conclusion of the paper.

Additionally, the entangle state is generated in a posteriori manner, and thus cannot propagate freely as a traveling field, which prohibits its applications in the context of quantum repeaters and quantum networks.

The demonstrated scheme would produce a freely propagating hybrid entangled resource if used in combination with heralded or on-demand sources of polarization entangled pairs of photons instead of a probabilistic one that is used in our work. Such sources do exist as we indicate in the article. Importantly, the use of such a source would also greatly reduce the probability of false positive events, which is the main reason for the fidelity reduction in our work. We add comments regarding this matter in the conclusion, as well as references [40,41] to on-demand entangled photon pair sources that augment the previously referenced papers [37-39] on heralded sources.

The success rate is intrinsically low in the reported experimental schemes, which not only limits the practical

usage, but also renders it impossible to implement fully tomography of the generated two-mode states. Due to the limited number of quadrature values (about 2500 for each state), the authors have to conduct the quantum state reconstruction in the subspace. As a result, the density matrices shown in Fig. 2 are not normalized. As far as I know, it is important to acquire sufficiently large amount of quadrature samples for quantum state reconstruction to reduce numerical errors in the maximum-likelihood algorithm. In the experiment, the correction of close to 1/2 detection efficiency in the maximum-likelihood algorithm may lead to a large numerical uncertainty for the reconstructed states, especially in the case of small amount of quadrature samples. Rigorously, the fidelities obtained in the paper should be given with corresponding uncertainties, particularly when they are compared with classical bounds.

We cannot agree that the number of points acquired is insufficient for tomography. We carefully kept track of the reconstruction errors by means of bootstrapping to make sure all the critical benchmarks are above the threshold level beyond the statistical uncertainty. In the new version of the paper, following the referee's advice, we included the uncertainties with all the fidelity figures. We also clarified our technique for reconstructing the state and evaluating the uncertainty in the Methods.

The fidelities presented in the text are corrected for a detection loss as large as 45%. I suppose that the raw fidelities without correction are not high enough to surpass the classical benchmark in the demonstration of entanglement and teleportation. To be rigorous, the classical bound for quantum witness should be calculated taking account of the detection efficiency. This is particularly important for continuous-variable non-classical states since they are notoriously fragile to losses.

Reconstructing the state with the correction for the detection efficiency yields the state of the electromagnetic mode *before* it has been detected. It is this state that has been produced as a result of the teleportation or remote preparation procedure, and hence must satisfy the appropriate benchmark to verify the quantum nature of that procedure. Therefore we believe that using the efficiency correction in reconstructing the state is justified.

For continuous-variable states, fidelity is a misleading figure of merit in some cases. For instance, it is commonly known that a state with a smaller energy is more robust to losses, thus easier to maintain a high fidelity with the original state. The classical bounds in this paper should depend on the size of coherent state as the basis in CV encoding.

While we cannot argue with the general statement made by the reviewer, we observe that the fidelity-based entanglement and quantum teleportation benchmarks derived in Refs. [26] and [34] are universal for *any* qubit or e-bit regardless of its physical nature. Because these benchmarks help us conclude the quantum nature of our procedures, their use as figures of merit in our work is justified.

Reviewers' comments:

Reviewer #1 (Remarks to the Author):

The authors have addressed many but not all the points raised by the referees. In addition, from my point of view, some answers are not satisfactory. In particular, the answers to the points raised by referees 1 and 3 are not so detailed and they lack of some important aspects. For instance, the problem connected to the small number of quadrature points used for tomography is only treated marginally. In fact, the problems connected to bias and data completeness in tomography are not taken into account [see Phys. Rev. A 95, 022107 (2017); Phys. Rev. A 86, 052123 (2012); Phys. Rev. A 89, 012305 (2014)]. Moreover, the bootstrap method used by the authors to estimate the fidelity errors is a good estimator for uncertainties in the limit of a high number of data when bias can be negligible. The question raised by referee 3 about normalization of density matrixes of Fig. 2 remains unsolved. In conclusion, considering the high-impact of Nature Communications, I would expect more care and clarifications in drawing up a reply letter. I am sorry but I cannot leave a positive comment after this round of review.

Reviewer #2 (Remarks to the Author):

Based on the revision of the manuscript and the reponses to all referees' comments, I stick to my initial assessment and recommend the paper for publication in Nature Communications. In particular, my own questions have all been addressed satisfactorily. The main criticism came from Referee 3, and also his/her comments have been addressed well. Especially, the important aspect that neither larger coherent-state amplitudes nor freely propagating hybrid entangled states are fundamentally ruled out with the present approach makes me conclude that the present concept and approach to optical hybrid QIP is sufficiently versatile and important to be presented to a broader audience.

Reviewer #3 (Remarks to the Author):

After careful reading of the response written by the authors, I decided to change my mind and to accept the paper for publication in the Nature Communications. My concern about the presented results is mainly the limited size of the generated cat state, which should be large enough for useful hybrid quantum information processing. Otherwise, the generated two-mode entangled states would be basically closed to DV-DV entangled states, which are merely detected and characterized in a hybrid way. Indeed, even if I remain expectation for more advanced results, I agree that the reported scheme is novel and interesting. I am satisfied with added paragraphs in the revised manuscript that explicitly discuss current limitations and give feasible solutions. In addition, I would like to highlight the degree of control that is required to generate these kind of complex states of light, which also played a role in my final decision.

=====

RESPONSE TO REFEREE #1

For instance, the problem connected to the small number of quadrature points used for tomography is only treated marginally. In fact, the problems connected to bias and data completeness in tomography are not taken into account [see Phys. Rev. A 95, 022107 (2017); Phys. Rev. A 86, 052123 (2012); Phys. Rev. A 89, 012305 (2014)].

According to the work Phys. Rev. A 86, 052123 (2012) mentioned by the reviewer, the number of quadrature measurements for a complete characterization of the state should be d^2 in the case of a d -dimensional space and $m > d$ different phases of the local oscillator must be used. The reconstruction subspace's dimension $d = 4$ in our case. In our case, we permitted the local oscillator phase to vary continuously and implemented no binning of the phases or quadratures. Thus, in our case, the formal criterion of information completeness is certainly satisfied. In the new version of a paper, we draw attention to this fact in the Methods.

Moreover, the bootstrap method used by the authors to estimate the fidelity errors is a good estimator for uncertainties in the limit of a high number of data when bias can be negligible.

In the new version of the work, we do take into account the systematic bias according to Phys. Rev. A 95, 022107 (2017) cited by the reviewer, as described in the Methods. The correction to the uncertainty due to the bias turned out to be negligible in all cases. We furthermore improved our bootstrapping technique, resulting, for some cases, in somewhat lower error estimates than in the previous version of the manuscript.

We have furthermore cited Phys. Rev. A 89, 012305 (2014) mentioned by the reviewer and presented an argument that the fidelity parameter is suitable for the purpose of our paper.

The question raised by referee 3 about normalization of density matrixes of Fig. 2 remains unsolved.

We clarify in the revised version that (subsection "Experiment: remote state preparation" and the Fig. 2 caption) that our reconstruction algorithm ensures that the reconstructed state in the CV channel is normalized. Only the projection of that state onto the subspace spanned by the positive and negative cats is not normalized; however, the deviation from the norm is relatively small. The only place where this unnormalized projection is used in the paper is Figs. 2 and 3.

We note that Reviewer #3, in their original report, has not asked any question regarding the normalization of density matrixes of Fig. 2; they simply stated the fact that the density matrices are not normalized. This is the reason why we have not commented on this in our original response.

=====

RESPONSE TO REFEREE #2

Based on the revision of the manuscript and the reponses to all referees' comments, I stick to my initial assessment and recommend the paper for publication in Nature

Communications. In particular, my own questions have all been addressed satisfactorily. The main criticism came from Referee 3, and also his/her comments have been addressed well. Especially, the important aspect that neither larger coherent-state amplitudes nor freely propagating hybrid entangled states are fundamentally ruled out with the present approach makes me conclude that the present concept and approach to optical hybrid QIP is sufficiently versatile and important to be presented to a broader audience.

We thank the reviewer for the positive evaluation of our work.

=====

RESPONSE TO REFEREE #3

After careful reading of the response written by the authors, I decided to change my mind and to accept the paper for publication in the Nature Communications. My concern about the presented results is mainly the limited size of the generated cat state, which should be large enough for useful hybrid quantum information processing. Otherwise, the generated two-mode entangled states would be basically closed to DV-DV entangled states, which are merely detected and characterized in a hybrid way. Indeed, even if I remain expectation for more advanced results, I agree that the reported scheme is novel and interesting. I am satisfied with added paragraphs in the revised manuscript that explicitly discuss current limitations and give feasible solutions. In addition, I would like to highlight the degree of control that is required to generate these kind of complex states of light, which also played a role in my final decision.

We thank the reviewer for the positive evaluation of our work.